# Forest Cover Change and the Effectiveness of Protected Areas in the Himalaya since 1998

**Changjun Gu** [1,2] , **Pei Zhao** [3], **Qiong Chen** [3], **Shicheng Li** [4] , **Lanhui Li** [5] , **Linshan Liu** [1,*] **and Yili Zhang** [1,2,6,*]

1  Key Laboratory of Land Surface Pattern and Simulation, Institute of Geographic Sciences and Natural Resources Research, CAS, Beijing 100101, China; gucj0308@gmail.com
2  University of Chinese Academy of Sciences, Beijing 100049, China
3  College of Geographic Sciences, Qinghai Normal University, Xining 810008, China; qhnuzp@163.com (P.Z.); qhchenqiong@163.com (Q.C.)
4  Department of Land Resource Management, School of Public Administration, China University of Geosciences, Wuhan 430074, China; lisc@cug.edu.cn
5  Fujian Key Laboratory of Pattern Recognition and Image Understanding, Xiamen University of Technology, Xiamen 361024, China; lilh.15b@igsnrr.ac.cn
6  CAS Center for Excellence in Tibetan Plateau Earth Sciences, Beijing 100101, China
*  Correspondence: liuls@igsnrr.ac.cn (L.L.); zhangyl@igsnrr.ac.cn (Y.Z.)

**Abstract:** Himalaya, a global biodiversity hotspot, has undergone considerable forest cover fluctuation in recent decades, and numerous protected areas (PAs) have been established to prohibit forest degradation there. However, the spatiotemporal characteristics of this forest cover change across the whole region are still unknown, as are the effectiveness of its PAs. Therefore, here, we first mapped the forest cover of Himalaya in 1998, 2008, and 2018 with high accuracy (>90%) using a random forest (RF) algorithm based on Google Earth Engine (GEE) platform. The propensity score matching (PSM) method was applied with eight control variables to balance the heterogeneity of land characteristics inside and outside PAs. The effectiveness of PAs in Himalaya was quantified based on matched samples. The results showed that the forest cover in Himalaya increased by 4983.65 km$^2$ from 1998 to 2008, but decreased by 4732.71 km$^2$ from 2008 to 2018. Further analysis revealed that deforestation and reforestation mainly occurred at the edge of forest tracts, with over 55% of forest fluctuation occurring below a 2000 m elevation. Forest cover changes in PAs of Himalaya were analyzed; these results indicated that about 56% of PAs had a decreasing trend from 1998 to 2018, including the Torsa (Ia PA), an area representative of the most natural conditions, which is strictly protected. Even so, as a whole, PAs in Himalaya played a positive role in halting deforestation.

**Keywords:** deforestation; fragmentation; Google Earth Engine; Himalaya; propensity score matching; protected areas; random forest

## 1. Introduction

Forests are vital for the global carbon cycle [1–3], biodiversity conservation [4–7], climate change [8,9], and, of course, human livelihoods [10,11]. According to the Global Forest Resources Assessment 2015 (FRA 2015) released by the Food and Agriculture Organization (FAO), which is located in the city center of Rome, the world's total forest area declined by 3% from 1990 to 2015 [12]. The study reportedly estimated a current global rate of forest loss at 0.6% per year [13]. Forest degradation poses a threat to biodiversity richness [14,15], ecosystem services [16–18], habitat quality [19,20], and invasive species [21,22], with forest cover loss and fragmentation regarded as the main causes of global ecosystem degradation [23]. Anthropogenic activities such as expansion of agriculture and

urbanism, as well as logging and burning of stands, are thought to be the primary causes of forest loss and fragmentation [24,25]. Under the dual action of climatic change and land-cover change, global biodiversity and ecosystem functioning are facing grave threats, especially in the world's recognized biodiversity hotspots [26,27].

Setting protected areas (PAs) is a well-known way to achieve forest conservation [28–30], and they have been established all around the world over the past decades [31]. According to the Protected Planet Report 2016, there are at least 202,467 PAs, together covering ca. 19.8 million km$^2$ [32]. Given the Aichi Biodiversity Target 11 of attaining "at least 17% of terrestrial and inland water areas and 10% of coastal and marine areas, especially areas of particular importance for biodiversity and ecosystem services", the area of PAs is expected to increase further. However, studies have found that forest degradation occurs near or even inside some PAs. From 1982 to 2000, approximately 70% of the surrounding buffers adjacent to PAs have suffered deforestation, while 25% of PAs experienced deforestation within their boundaries [33]. Deforestation near and inside PAs is considered as one of the main causes aggravating the ecological services of these biodiversity arks [34]. Hence, the effectiveness of PAs in protecting forest cover is noteworthy. Nevertheless, the effectiveness of the PAs remains controversial [35,36], including their level of forest protection [37]. Although some research indicated that PAs' establishment could effectively reduce deforestation and degradation [38], other work suggests that they contribute little to forest protection [39], with one study pointing out that the effectiveness of PAs for forest conservation may be overestimated [37]. Accordingly, the development of PAs should shift from quantity to quality, which entails the effectiveness of existing PAs as a necessary step [40].

The Himalaya region is among the 36 recognized global biodiversity hotspots [41], and it has experienced forest degradation in recent decades [42]. Pandit et al. (2007) [43] reported on an alarming trend of deforestation in the Indian Himalaya and projected the likely consequential extinctions of endemic taxa (species and subspecies) by 2100 across a broad range of taxonomic groups, including gymnosperms, angiosperms, fishes, amphibians, reptiles, birds, and mammals. Kanade et al. (2018) [27] assessed the influence of deforestation and degradation in the Sikkim Himalaya in India, finding a 16% decline in primary forest cover between 1990 and 2013. Chakraborty et al. (2017) [44], in characterizing fragmentation trends of the Himalayan forests in the Kumaon region of Uttarakhand, India, pointed out that the region was undergoing intensive forest fragmentation. Earlier, Joshi et al. (2014) [45] identified spatial trends of forest degradation from 1979 to 2009; this revealed that forested areas were subject to degradation and isolation due to the loss of connecting forest stands. However, since most research has mainly focused on trends at regional scales, such as Western Himalaya [46], Central Himalaya [47–50], and Eastern Himalaya [51], there is little research analyzing forest cover change across the whole Himalaya, and those carried out have used different temporal and spatial scales [17]. Currently, Himalaya has 172 PAs, totaling 109,975.19 km$^2$, or 18.5% of its area, but the effectiveness of protected area management there is still facing formidable challenges [52]. The current lack of relevant research has limited our perceptions about the effectiveness of these PAs and their better management.

Many studies have detected the changes in forest cover and monitored the forest fragmentation by utilizing remote sensing data and skills at regional and global scales [53–55]. Landsat data, with its long time series, high spatial resolution, and free access, has been extensively used in forest cover change detection [56–58]. The emergence of the cloud-based computing platform Google Earth Engine (GEE), with its strong calculation and storage capacities, has attracted broad attention [59] and has been widely applied in vegetation monitoring [60,61], crop mapping [62,63], and land cover classification [64,65]. The Landsat data archived on the GEE platform provides a unique opportunity to monitor forest cover change at high spatial resolutions, from local to global scales.

Given the above, this study's objective was to address the following questions: (1) What are the forest disturbance patterns in Himalaya from 1998 to 2018? (2) How do these patterns vary across time and along elevation? (3) Are the PAs in Himalaya effectively deterring forest cover loss?

## 2. Materials and Methods

The overall methodological framework for this study is depicted in Figure 1. First, based on the Landsat Surface Reflectance and a 70% training dataset, a random forest supervised classification on GEE was performed. The remaining 30% training dataset was used to assess the performance of our model. We mapped high-accuracy forest cover present in the years 1998, 2008, and 2018. Then, the landscape fragmentation tool (LFT v2.0) was applied to detect the forest fragmentation patterns during the study period. The 'spatial analyst module' in ArcGIS v10.4.1 was applied to determine changes in forest cover and fragmentation. Overlapping the PAs' boundaries and forest cover data, the effectiveness of PAs in mitigating forest cover loss was further assessed based on propensity score matching (PSM) skills.

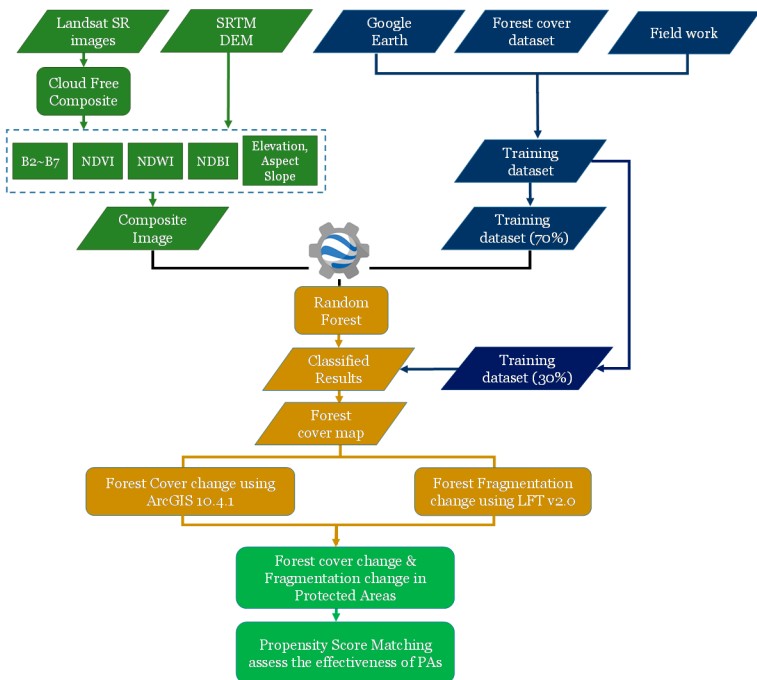

**Figure 1.** The methodological framework of this study.

### 2.1. Study Area

The study area is located around the southern edge of the Tibetan Plateau [66] and the north of the Indian subcontinent, covering the entire Himalaya (Figure 2). The Himalaya mountain range spans five countries, including Bhutan, China, India, Nepal, and Pakistan, extending ca. 2400 km from its northwest to southeast, covering an estimated area of $6.5 \times 10^5$ km$^2$ [67]. Great variation in climate and vegetation properties can be found here because of the elevation, which ranges from 50 m in the Southern Himalayan lowlands up to 8844 m (Mt. Everest). This generates climatic gradients and soil types [68]. The southern aspect receives abundant rainfall and features lush vegetation, except in its high elevation areas, while the northern slope gets little rainfall and has sparse vegetation due to the influence of topography and atmospheric circulation. With greater elevation, the temperature and precipitation in the mountain area also change accordingly. These pronounced spatial variations result in a diversity of ecosystems. Himalaya is a unique region in terms of land cover (featuring the ice, grass, shrub, forest, and farmland, among others) and is rich in biodiversity. This study focused on characterizing forest fragmentation dynamics in the Himalayan region during the past 20 years.

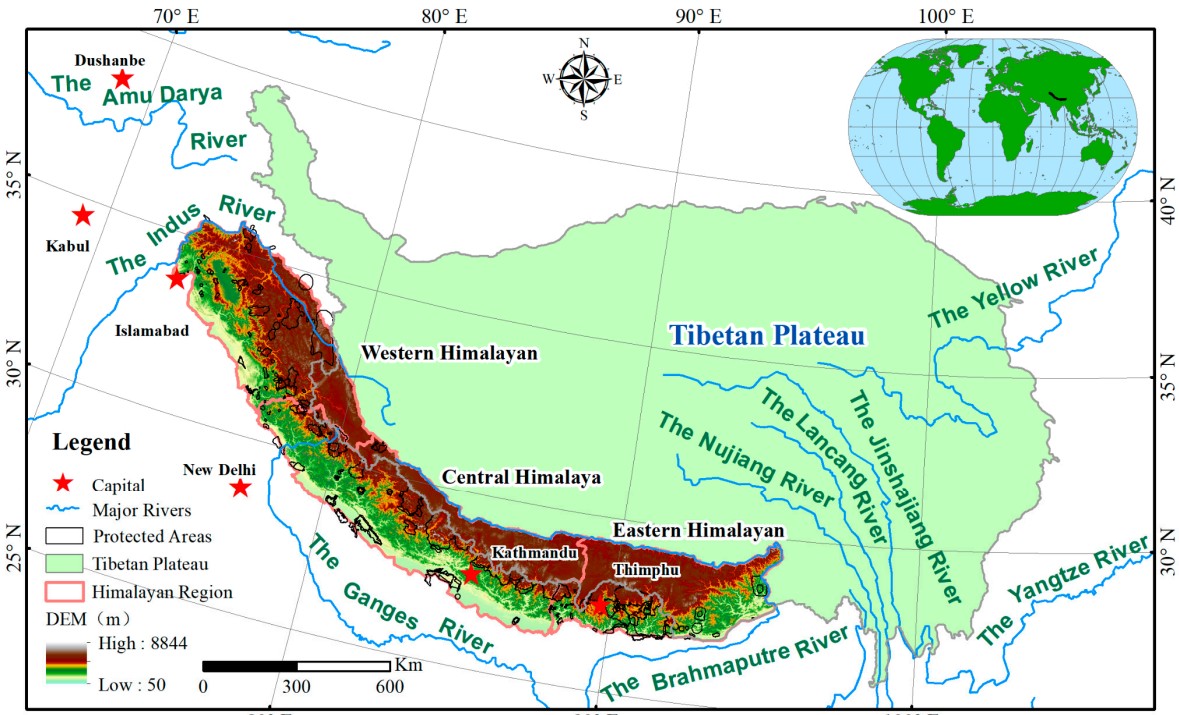

**Figure 2.** Location of the study area. It spreads across Bhutan, China, India, Nepal, and Pakistan. The Himalaya boundary and Tibetan Plateau boundary come from Nie et al. (2017) and Zhang et al. (2014), respectively. The protected area (PA) boundary was obtained from https://www.protectedplanet.net/c/wdpa-lookup-tables. The Tibetan Plateau boundary is available at http://www.geodoi.ac.cn/WebEn/doi.aspx?doi=10.3974/geodb.2014.01.12.v1.

### 2.2. Satellite Imagery and Pre-Processing of Training Data

Atmospherically corrected Landsat Surface Reflectance products (30 m resolution) were used in this study [69,70]. Landsat, with its long time series and high spatial resolution, has been widely used for forest monitoring [71,72], land cover classification [73–75], water body extraction [76,77], and crop mapping [78]. The Landsat-5 Thematic Mapper (TM) Surface Reflectance images were acquired in 1998 and 2008; Landsat-8 SR images were acquired in 2018. Cloud-free images were critical but difficult to obtain for the Himalaya region due to its dense cloud cover. So, a two-step method was applied here. First, we selected all the available images in the whole year to yield a multi-image composite. Then, we employed the C Function of Mask (CFMASK) algorithm to mask the clouds and shadows based on the pixel_qa band [79]. The whole approach was developed in the GEE [59]. Terrain factors (SRTM: http://srtm.csi.cgiar.org/srtmdata/, 90 m resolution, elevation, aspect, and slope were selected) were also taken as ancillary data to achieve higher classification accuracy.

Training data were obtained from our field work, high-resolution Google Earth images, and other forest cover productions (Figure 3). Our research team had carried out field investigations in both 2018 and 2019. In particular, the quadrat-based survey, field photos, and drone flights were applied to record information on land cover. By comparing the field-recorded data with Landsat images, much experience and knowledge were obtained. Given the long-spanning mountain range of Himalaya, high-resolution Google Earth imagery was referenced to select training data points based on prior knowledge. To avoid spatial correlation, random points were created in the Himalaya region, and only points that were recognized as forest points were kept in the GEE. We also sampled training points from the existing forest cover datasets by using the method described in Hu et al. (2019) [80].

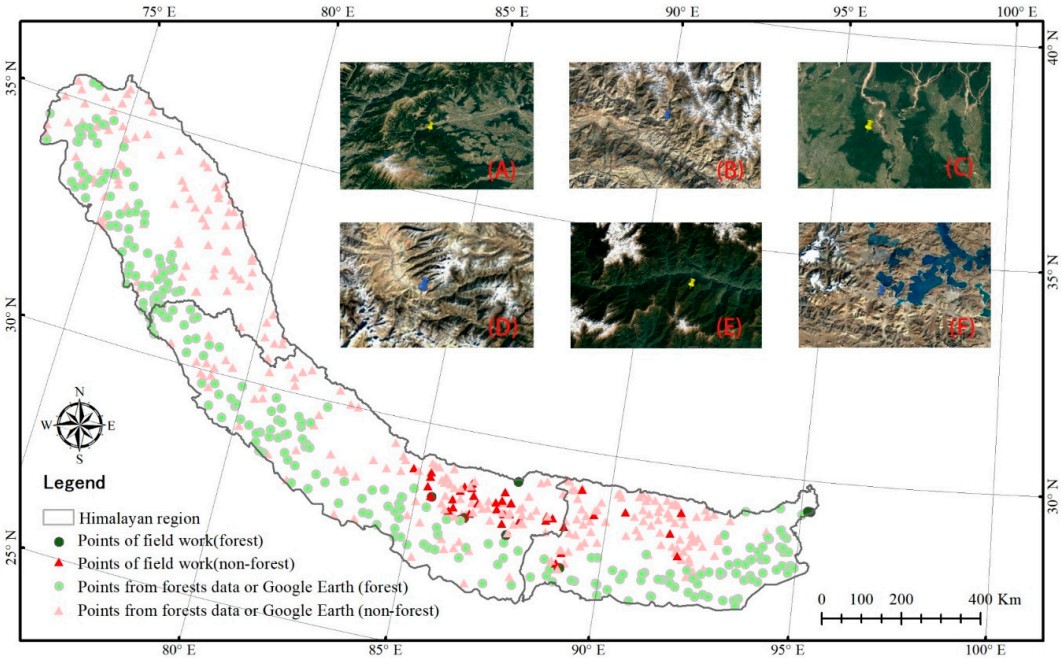

**Figure 3.** A total of 600 units of training data were collected from the field work in 2018 and 2019, Google Earth, and other forest cover data. (**A**) Forest point (Western Himalaya), (**B**) non-forest point (Western Himalaya), (**C**) forest point (Central Himalaya), (**D**) non-forest point (Central Himalaya), (**E**) forest point (Eastern Himalaya), and (**F**) non-forest point (Eastern Himalaya).

### 2.3. Input Features and Classification

Spectral indexes have been widely used in land cover classification [81–83]. The spectral bands of Landsat Surface Reflectance images were selected as the main input features. Furthermore, we also chose the Normalized Difference Vegetation Index (NDVI) [84], the Normalized Difference Built Index (NDBI) [85], and the Normalized Difference Water Index (NDWI) [86] to improve the classification performance (Equations (1)–(3)). As mentioned above, terrain factors were also incorporated into the classification.

$$\text{NDVI} = \frac{\rho\text{NIR} - \rho\text{Red}}{\rho\text{NIR} + \rho\text{Red}} \tag{1}$$

$$\text{NDWI} = \frac{\rho\text{Green} - \rho\text{NIR}}{\rho\text{Green} + \rho\text{NIR}} \tag{2}$$

$$\text{NDBI} = \frac{\rho\text{MIR} - \rho\text{NIR}}{\rho\text{MIR} + \rho\text{NIR}} \tag{3}$$

where $\rho$NIR, $\rho$Red, $\rho$Green, and $\rho$MIR in the equations above represent the surface reflectance values of the Near-Infrared band (0.76–0.9 μm), Red band (0.63–0.69 μm), Green band (0.52–0.6 μm), and Mid-Infrared band (2.08–2.35 μm).

The random forest (RF) was selected to classify forest land cover classification because it is capable of handling multisource environmental variables, providing an ensemble of multiple decision-tree-type classifications, and also performing well when compared with other similar algorithms for land cover classification [87,88]. Importantly, this method also models interactions and nonlinear relationships among environmental variables and performs well when interpolations are required [89]. Not surprisingly, RF is now widely used in land cover mapping [90,91]. More specifically, RF classification is a relatively well-known supervised machine learning algorithm that iteratively produces an ensemble of decision tree classifications by using corresponding randomly selected subsets of the training dataset [89]. It grows classification trees by splitting each node using

a random selection subset of input variables, which reduces overfitting and yields a more robust classification compared to other classifiers [89].

## 2.4. Detection of Forest Fragmentation

We first used forest cover maps in 1998, 2008, and 2018 to detect the dynamics of forest fragmentation. Then, we computed the annual rate of forest cover change and mapped the regions characterized by forest loss or forest gain. The ArcGIS landscape fragmentation tool (LFT v2.0) was utilized to identify the forest fragmentation and edge effects [92]. This method first reclassifies the forest cover pixels and non-forest pixels as 1 and, 0, respectively; the non-forest land cover types were presumed to be the cause of fragmentation. By computing the distance of forest pixels to non-forest pixels, a classification of fragmentation can be built, entailing six classes of forest: Patch, edge, perforated, small core (SC) (<250 acres), medium core (MC) (250–500 acres), and large core (LC) (>500 acres) (Figure 4). The core forest corresponds to any forest pixels at least 100 m from the nearest non-forest pixel. Patch pixels are within small forest fragments that do not contain any core forest pixels. Perforated and edge forests are with 100 m of non-forest pixels, but are still part of a tract containing forest core pixels. Edge pixels lie along the outside edge of the forest tract, while perforated pixels occur along the edge of small forest canopy gaps. Here, the edge width was defined as 100 m, following a previous study [16].

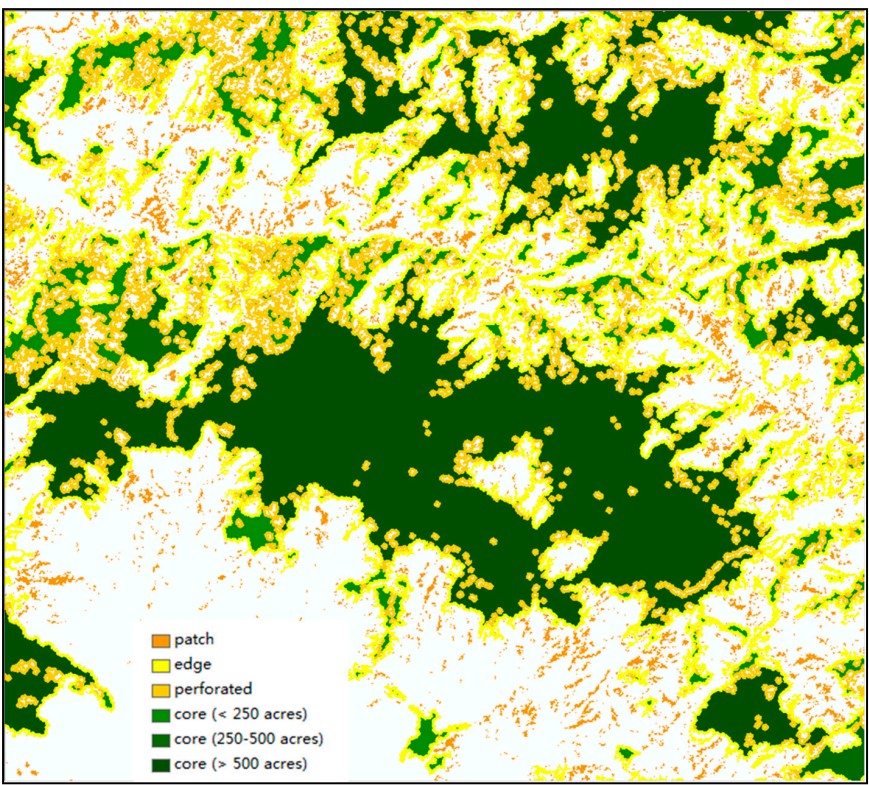

**Figure 4.** The demonstration of forest fragmentation types.

## 2.5. Effectiveness Assessment of PAs

Oestreicher et al. [93] defined the effectiveness of a given PA as a function of the external pressures exerted on a PA. Other research also applied a similar theory to assess the effectiveness of global PAs in resisting forms of anthropogenic pressures [94]. Accordingly, here, we adopted the deforestation rates in three periods (i.e., 1998–2008, 2008–2018, and 1998–2018) as an indicator of pressures operating beyond PAs. Based on this definition, we screened and analyzed existing PAs of Himalaya. First, we ignored PAs lacking forest cover, and, for comparative purposes, the PAs built at unknown times

were also excluded. According to the classification system defined by the International Union for the Conservation of Nature (IUCN), which is located in Gland, Switzerland, there are seven kinds of PAs based on seven management categories (Ia, Ib, II, III IV, V, VI). Category Ia includes strictly protected areas representative of the most natural conditions [95]. Those PAs defined as 'Not Reported', 'Not Applicable', or 'Not Assigned' were also excluded from our study. For the first period (1998–2008), only PAs established in 1998 or earlier (a total of 88 PAs) were taken into consideration; for the second period (2008–2018), PAs established in 2008 or earlier were selected. For statistical convenience, the lone PA established in 2009 and the four PAs established in 2010 were also included in the second period (for a total of 108 PAs). In this way, we had 108 PAs in total (Ia: 1, Ib: 0, II: 24, IV: 56, V: 1, VI: 26; Table S1).

Previous studies assessed the effectiveness of PAs based on the assumption that the PAs and their surrounding areas are spatially homogeneous. Accordingly, the buffer zone was built around the PAs, and their inside and outside were compared for differences [96]. However, spatial heterogeneity can hardly be ignored, as doing so may bias this kind of assessment [97]. More importantly, the locations of PAs are not randomly set, are often biased towards remote areas, and may have a higher elevation, steeper slope, and lower suitability for agriculture [98,99]. This means that even in the absence of protection, the pressures on these areas are expected to remain low [94,100]. Here, propensity score matching (PSM) was conducted to assess what would have happened if a protection had never been applied in these areas (Figure 5) [101].

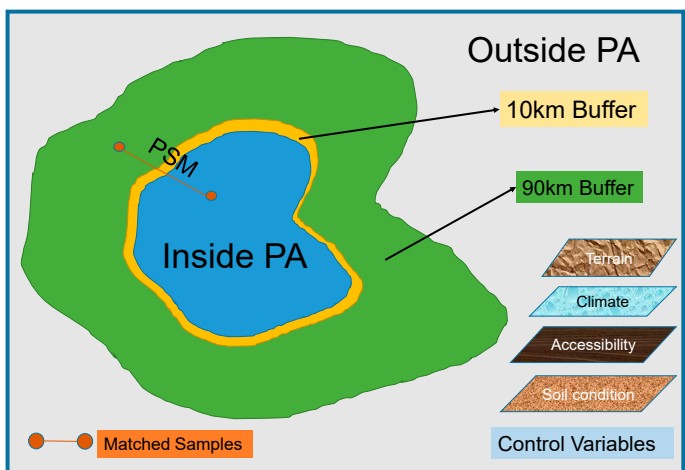

**Figure 5.** The Demonstration of propensity score matching (PSM). Eight control variables were selected, including terrain conditions (elevation, slope), climate (precipitation, temperature), human activities (distance to roads, rivers, and settlements), and soil conditions (soil organic carbon).

To do this, first, a 10 km buffer was built outside the PAs, and samples inside this buffer would not be selected to avoid the spillover effects of PAs [102]. Then, a 90 km buffer was set near the 10 km buffer and defined as an unprotected area (i.e., outside PAs). We randomly sampled ca. 10,000 pixels inside (setting their value as 1) and outside PAs (setting their value as 0). To avoid any spatial autocorrelation, we set the closest distance between any two pixels to be no less than 300 m [103]. Terrain, climate, soil, and accessibility have all been found to be statistical factors influencing the forest loss [99,104]. Then, based on eight selected control variables—elevation, slope, soil organic carbon, annual precipitation, annual average temperature, distance to the nearest major road, distance to the nearest major settlement, and distance to the nearest river (Table S2)—the PSM algorithm calculated the propensity score for each of the sample pixels inside and outside PAs, such that pixels with the most similar propensity scores were matched [105,106]. The main parameters were set as follows: The 'nearest' method was selected and the argument 'ratio' was set to 1 to ensure that an 'apples to apples' comparison was made. A caliper of 0.01 was set as the maximum acceptable distance for matching. PSM was implemented using the MatchIt package in R v 3.6.1 [107].

For inference, we relied on the absolute effects (AEs) and relative effects (REs), as described by Zhao et al. (2019) [100]. The absolute effects (AEs) and relative effects (REs) of PAs were calculated as follows: The AE of PAs was the numerical difference between the deforestation rate inside PAs and their matched outside PAs, expressed as AE = OutsideDef − InsideDef, where OutsideDef (InsideDef) was the percentage of outside (inside) PAs' sample pixels where forest loss occurred out of all matched pixels. A positive AE would indicate that PAs effectively protected against deforestation. The RE measured how far the baseline deforestation rate had been altered by the PAs' existence.

$$\text{AE} = Outside_{Def} - Inside_{Def} \tag{4}$$

$$\text{RE} = \left(Outside_{Def} - Inside_{Def}\right)/Outside_{Def} \tag{5}$$

where the $Inside_{Def}$ denotes the deforestation rate inside PAs and the $Outside_{Def}$ denotes the deforestation rate outside the PAs.

## 3. Results

### 3.1. Accuracy Assessment

The overall accuracy of 1998, 2008, and 2018 was at least 94% with a kappa coefficient higher than 89%. The overall accuracy and kappa coefficient were highest, at 96.19% and 92.38%, in 2018, and were lowest in 1998, at 94.76% and 89.42%, respectively. For a better visual assessment of our results, three cities were selected; a regional amplification map is depicted in Figure 6. We then compared the high-resolution Google Earth imagery to our results; this showed that the latter were able to accurately depict forest distributions on complicated terrain.

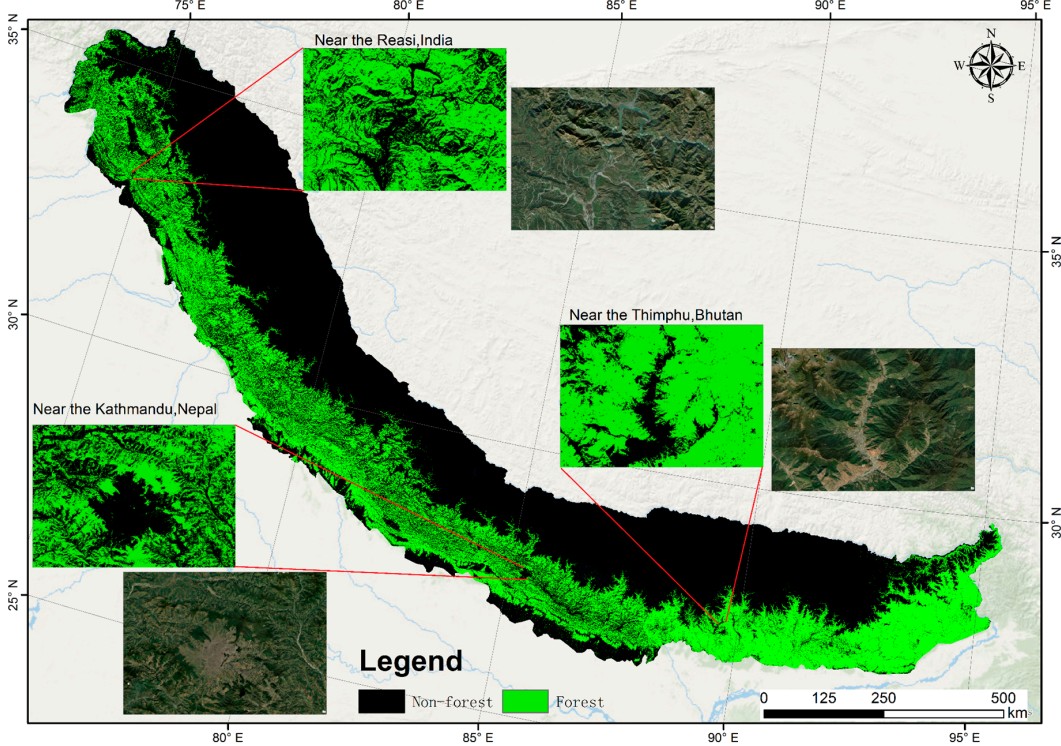

**Figure 6.** Demonstration of the classification results for the year 2018. Three cities in Western Himalaya (Reasi in India), Central Himalaya (Kathmandu in Nepal), and Eastern Himalaya (Thimphu in Bhutan) were selected.

*3.2. Forest Cover Changes Over the Past 20 Years*

The forest cover maps in 1998, 2008, and 2018 are shown in Figure S1, for which their respective total areas of forest cover were 209,912.79 km$^2$, 214,896.44 km$^2$, and 210,163.73 km$^2$, respectively (Table 1). For the entire study period, forest cover in Western Himalaya (WH) showed a decreasing trend, while in Central Himalaya (CH) and Eastern Himalaya (EH), it showed a modest increase. From 1998 to 2008, there was a net increase in forest cover in WH, CH, and EH; the largest gain was in EH, amounting to 3175.77 km$^2$, whereas forest cover increased only slightly in WH. By contrast, from 2008 to 2018, all three regions declined in forest cover, but the largest reduction occurred in EH (79,939.49 km$^2$ down to 76,787.10 km$^2$; Table 1).

**Table 1.** Forest cover of Himalaya (Western Himalaya (WH), Central Himalaya (CH), and Eastern Himalaya (EH)) in 1998, 2008, and 2018.

| Forest Cover (km$^2$) | 1998 | 2008 | 2018 | Change 1998–2008 | Change 2008–2018 | Change 1998–2018 |
|---|---|---|---|---|---|---|
| Western Himalaya (WH) | 43,643.61 | 44,194.35 | 42,910.37 | 550.74 | −1283.99 | −733.25 |
| Central Himalaya (CH) | 89,505.46 | 90,762.60 | 90,466.27 | 1257.15 | −296.33 | 960.81 |
| Eastern Himalaya (EH) | 76,763.72 | 79,939.49 | 76,787.10 | 3175.77 | −3152.39 | 23.38 |

We obtained another statistic concerning the deforestation and reforestation along the elevation (Figure S2, Table 2). These results showed that, from 1998 to 2018, deforestation and reforestation mainly occurred in regions lying below an elevation of 2000 m, which accounted for more than 55% of total forest losses and gains. The same pattern was also found for the 2008–2018 period. From 1998 to 2008, the largest proportion of land corresponding to deforestation (15.01%) and reforestation (16.23%) was found at an elevation between 500 and 1000 m, and likewise from 2008 to 2018 (Table 2). In short, the changes in forest in Himalaya mainly occurred in lower-lying areas (>2000 m elevation), where deforestation and reforestation predominated between 500 and 1000 m during the 20-year period.

**Table 2.** Reforestation and deforestation levels along the elevation of Himalaya in 1998–2008, 2008–2018, and 1998–2018.

| Elevation (m) | 1998–2008 (%) | | 2008–2018 (%) | | 1998–2018 (%) | |
|---|---|---|---|---|---|---|
| | Reforestation | Deforestation | Reforestation | Deforestation | Reforestation | Deforestation |
| <500 | 12.12 | 14.27 | 12.26 | 13.20 | 14.78 | 12.25 |
| 500–1000 | 16.23 | 15.01 | 14.89 | 16.42 | 16.13 | 16.01 |
| 1000–1500 | 14.85 | 13.14 | 13.28 | 15.26 | 15.03 | 14.92 |
| 1500–2000 | 13.59 | 13.31 | 14.74 | 12.91 | 13.15 | 14.74 |
| 2000–2500 | 8.12 | 7.76 | 7.73 | 6.61 | 7.43 | 8.77 |
| 2500–3000 | 8.93 | 8.24 | 7.31 | 7.72 | 8.16 | 8.68 |
| 3000–3500 | 12.81 | 12.24 | 11.90 | 14.47 | 12.61 | 10.98 |
| 3500–4000 | 9.58 | 9.45 | 9.85 | 9.41 | 8.08 | 8.53 |
| >4000 | 3.76 | 6.59 | 8.03 | 3.99 | 4.63 | 5.11 |

During the 20-year period, core—small core (SC), medium core (MC), and large core (LC)—was the dominant forest type, accounting for above 50% of the total forest cover, followed by perforated and edges, whereas the patch type was the least prevalent (Figure 7). The core type accounted for 56.58% of the PAs' total area in 1998 and increased to 59.34% in 2008. From 2008 to 2018, this number decreased to 56.08%. The LC increased from 50.33% in 1998 to 53.59% in 2008, and then decreased to 49.71% in 2018, while the opposite trend was found for SC and MC, whose proportions decreased from 1998 to 2008, but then increased from 2008 to 2018. Similar trends were also observed for the patch, edge, and perforated types.

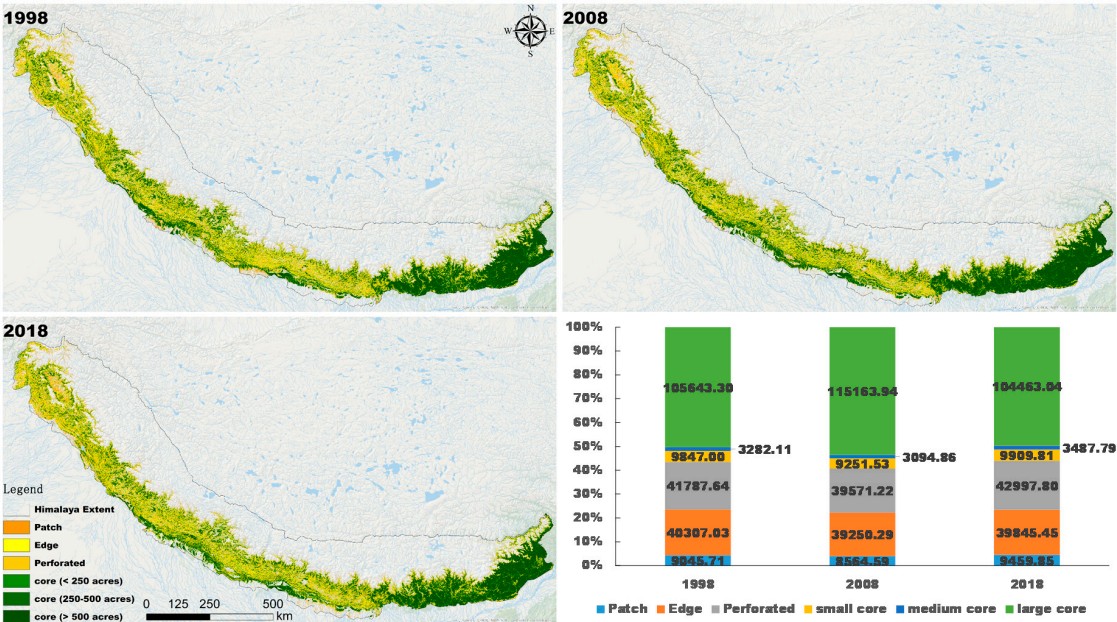

**Figure 7.** Changes in forest fragmentation patterns of Himalaya from 1998 to 2018.

The LC was distributed mainly in EH, where it accounted for >77% of the total LC in Himalaya during the study period, followed by CH (Figure 8). Edge and perforated types were each distributed primarily in WH, together amounting to >50%. From 1998 to 2018, LC in EH increased from 77.93% in 1998 to 81.92% in 2008, but then decreased to 77.20% in 2018. A similar trend was found in CH, where LC increased from 37.63% in 1998 to 41.34% in 2008, and then decreased to 38.46% in 2018. Both edge and perforated types had opposite patterns in CH vs. EH. In WH, edge increased from 27.76% in 1998 to 27.96% in 2008, and then decreased to 27.14%, while perforated continually increased over the 20-year period; however, in LC, perforated declined from 1998 to 2018.

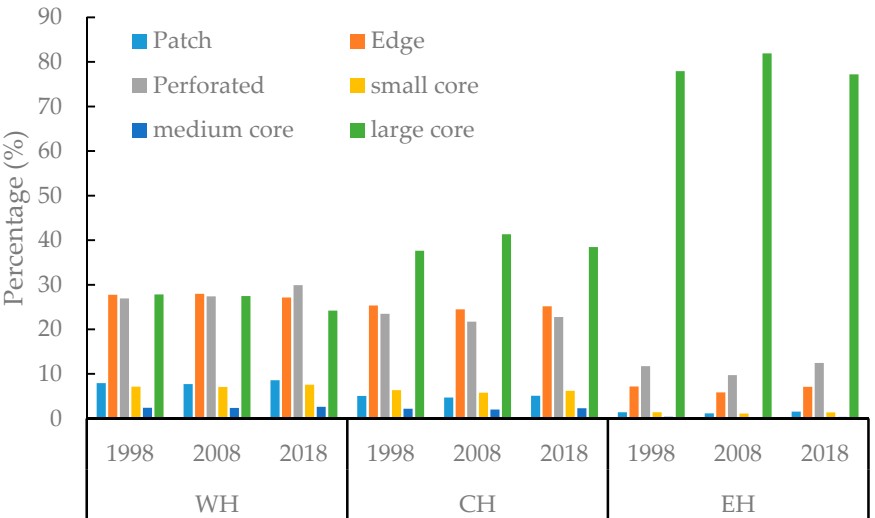

**Figure 8.** Forest fragmentation patterns in Western Himalaya (WH), Eastern Himalaya (EH), and Central Himalaya (CH) in 1998, 2008, and 2018.

We used a Sankey diagram to visualize the forest fragmentation changes in Himalaya from 1998 to 2018 (Figure 9). In Figure 9, the dark gray bars represent the non-forest (NF) in 1998, 2008, and 2018. The six stacked bars differing in color are used to visualize the six categories of forest cover in 1998, 2008, and 2018. The height of each bar corresponds to the proportion of total forest cover.

The transition lines between any two given years indicate the transfers from one type to another type, for which line width is proportional to the amount of converted land area.

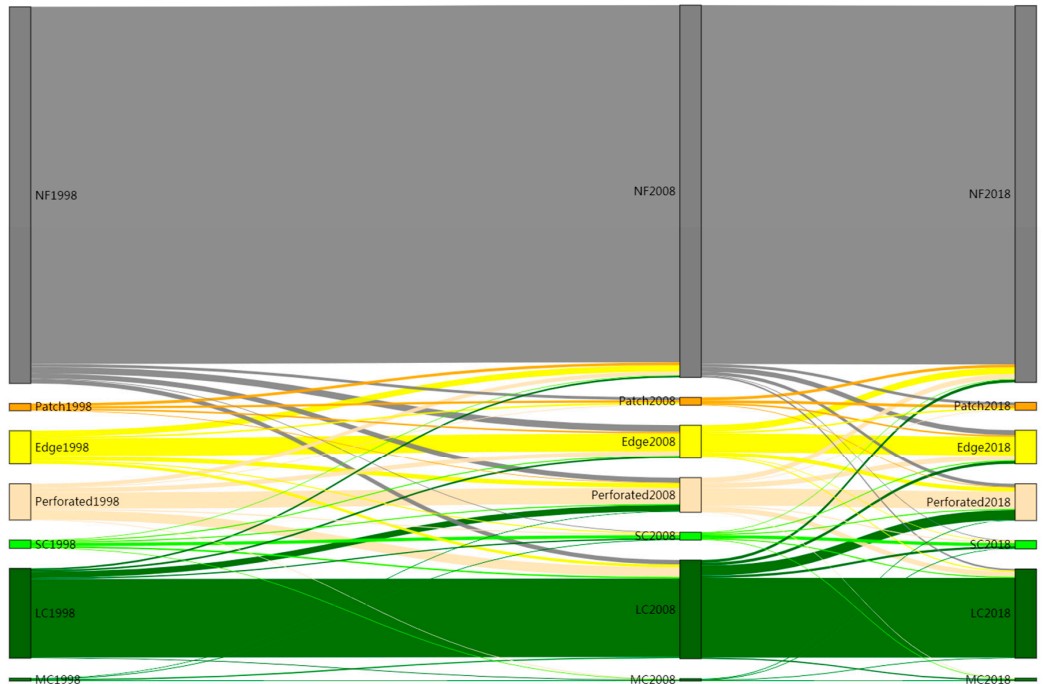

**Figure 9.** Sankey diagram for comparison of forest fragmentation dynamics created using the Sankey Diagram Generator (https://sankey.csaladen.es). Patch1998 refers to the Patch in 1998, and the rest can be interpreted in the same manner. NF refers to non-forest.

The conversion of different forest types can be easily interpretable from Figure 9. The previous analysis showed that forest cover increased, overall, from 1998 to 2018, and then decreased from 2008 to 2018. From 1998 to 2018, the decrease of non-forest arose mainly from conversion to edge (1.21%); conversely, from 2008 to 2018, the increase of non-forest came from converted edge (1.16%). The cores (SC, MC, and LC) are the predominant forest landscape type in all three years, with LC present in the highest proportion. From 1998 to 2008, the decrease of LC was mainly driven by its conversion to the perforated type (1.09%), which often occurred inside the LC; conversion from LC to edge was a secondary driver (0.30%), which occurred in the edge of the LC. Meanwhile, the transfer from perforated to LC contributed most to the increase of LC (1.68%), but the transfer from edge to LC contributed another 0.76%. The decrease of edge mainly happened via transferal to non-forest (1.1%), perforated (0.84%), and LC (0.52%). The decrease of SC mainly consisted of its transfer to LC (0.34%) and perforated (0.22%) types. Both MC and patch represented just a small proportion all forest types, and were mainly transferred to LC and edge, respectively. From 2008 to 2018, the conversion from LC to perforated (1.78%) was the principal cause of LC's decrease, followed by non-forest (0.48%) and edge (0.47%). The decrease of perforated was driven by transfers to LC (0.89%). The decrease of edge mainly resulted from its conversion to non-forest (1.16%). The decrease of SC corresponded to transfers to the perforated (0.24%), LC (0.18%), and edge (0.17%) types. The decreases of patch and MC were mainly due to their transfer to non-forest (0.52%) and edge (0.25%), respectively.

In summary, the forest cover in Himalaya has fluctuated over the past 20 years; having first increased and then decreased, the forest cover in Himalaya increased overall by 250.94 km$^2$ from 1998 to 2018, and the loss of forest cover mainly occurred in the 10 most recent years. The edge and perforated types showed an inverse trend. From the perspective of fragmentation, cores in Himalaya increased initially from 1998 to 2008, after which they decreased from 2008 to 2018. Edge converted into non-forest and non-forest converted into edge during the two time periods contributed most to

the temporal variation in forest cover. Hence, the fluctuation of forest cover mainly occurred at the edges of forest stands.

### 3.3. Effectiveness of PAs in Halting the Deforestation

The forest cover changes in the PAs of Himalaya were analyzed in three study periods (Figure 10). From 1998 to 2018, forest cover in 53 PAs showed a decreasing trend, even in the sole Ia PA (Torsa), for which the average loss was 3.91%. Five of these PAs were at the category II level, while 37 PAs and three PAs were at the IV and VI levels, respectively. Among these PAs, the greatest decrease was found in the Chitwan Buffer Zone (VI), at 13.04%; however, Chitwan (II) displayed an inverse trend, increasing by 1.75% from 1998 to 2018.

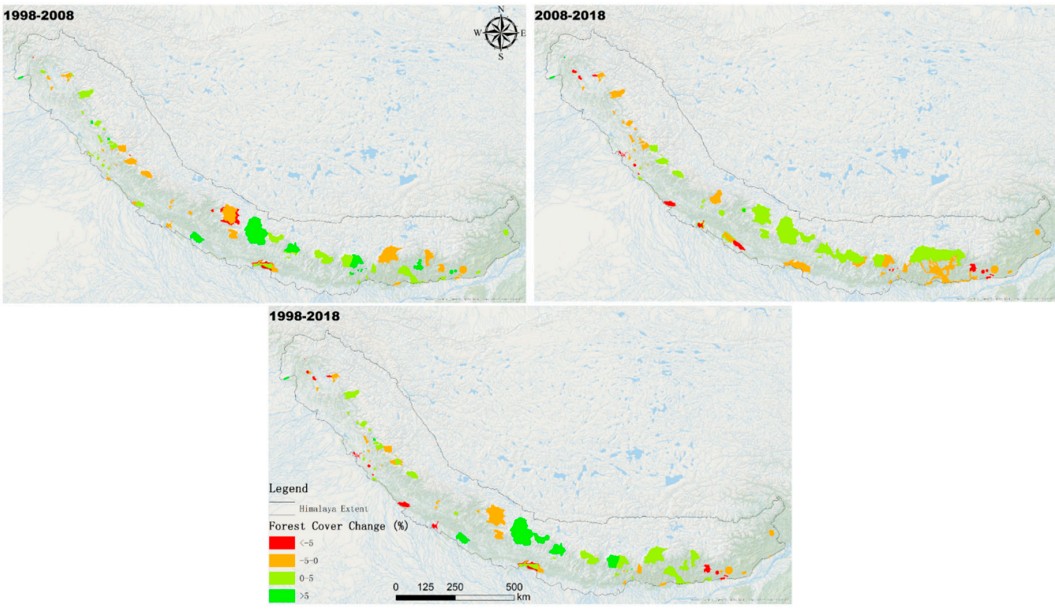

**Figure 10.** Forest cover change in the PAs during the study period.

During the 20-year period, forest cover in 42 PAs showed an increasing trend, with an average gain of 3.84%; 17 of those PAs were at the II level, while 19 PAs were at the IV level, one PA was at the V level, and five PAs were at the VI level. The greatest decrease was found in Biological Corridor 6. Based on the PSM, a total of 5297 pixels inside and outside 88 PAs were successfully matched from 1998 to 2018 (Table 3, Figure 11). The *p*-values of all control variables were greater than 0.05, thus indicating no significant difference between two matched groups. This confirmed that the inside pixels were successfully matched with the outside pixels. From 1998 to 2018, the deforestation rate inside the PAs was 4.83%, while it was 5.34% outside the PAs. The AE and RE of PAs from 1998 to 2008 were 0.51% and 9.55%, respectively.

**Table 3.** Balance of control covariates from 1998 to 2008. Values are the mean (SD).

| Control Variables | Inside PAs (1) | Outside PAs (0) | *p*-Value |
|---|---|---|---|
| | *n* = 5297 | *n* = 5297 | |
| DistanceToRiver | 1862.03 (1279.29) | 1883.82 (1296.71) | 0.385 |
| DistanceToRoad | 5747.56 (5494.43) | 5886.26 (5609.51) | 0.200 |
| DistanceToSettlement | 29,118.47 (19,869.79) | 29,421.12 (21,581.90) | 0.454 |
| Elevation | 2255.05 (1551.45) | 2292.32 (1552.62) | 0.218 |
| Precipitation | 1355.09 (742.46) | 1360.84 (700.17) | 0.683 |
| Slope | 23.04 (12.61) | 22.80 (13.31) | 0.339 |
| Soil organic carbon | 3.94 (3.10) | 4.03 (2.61) | 0.113 |
| Temperature | 12.45 (9.33) | 12.20 (9.36) | 0.161 |

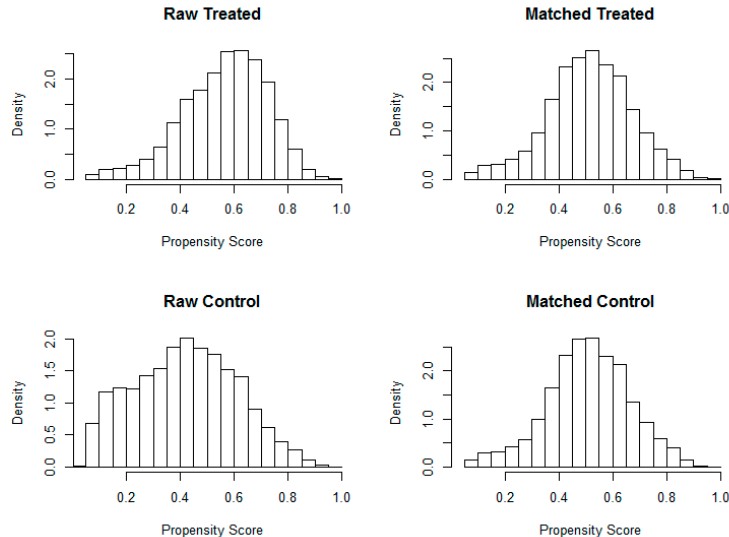

**Figure 11.** Distribution of propensity scores before and after propensity score matching (PSM) from 1998 to 2008. The Raw Control panel indicates the randomly selected pixels outside the PAs, and the Raw Treated panel indicates randomly selected pixels inside the PAs. The Matched Control panel denotes the matched pixels outside the PAs, while the Matched Treated panel denotes the matched pixels inside the PAs.

From 1998 to 2008—a total of 88 PAs were established before 1998 (1 Ia PA, 22 II PAs, 56 IV PAs, 1 V PA, and 8 VI PAs were included)—forest cover in 35 PAs showed a decreasing trend, with an average loss of 2.63%. Six of these PAs were at the II level, while 26 PAs and three PAs were at the IV and VI level, respectively. The largest decrease (10.30%) was observed in the Rara Buffer Zone. In all, 67 PAs showed an increasing trend from 1998 to 2008, with the greatest gain (23.03%) occurring in the Kyongnosla Alpine region; just one of these PAs was at the Ia level, while 16 PAs were at the II level, 30 PAs were at the IV level, one PA was at the V level, and five PAs were at VI level. From 1998 to 2008, a total of 5297 pixels were successfully matched. The $p$-values of all control variables were greater than 0.05, indicating there was no significant difference between two matched groups. This confirmed that the inside pixels were successfully matched with the outside pixels. From 1998 to 2008, the deforestation rate inside PAs was 3.3%, while it was 3.76% outside the PAs. The AE and RE of PAs from 1998 to 2008 were 0.46% and 12.23%, respectively.

From 2008 to 2018, 73 PAs showed a decreasing trend, with an average loss of 3.53%. As before, one PA was at Ia level, but 14 PAs were at the II level, 44 PAs were at the IV level, and 14 PAs were at the VI level. The greatest decrease happened in Eagle Nest, at 15.59%. By contrast, 35 PAs showed an increasing trend, with an average gain of 2.91%. The largest increase was observed in Manshi; 10 PAs were at the II level, while 12 PAs were at the IV level, one PA was at the V level, and 12 PAs were at the VI level. From 1998 to 2008, a total of 5297 pixels were successfully matched. The $p$-values of all control variables were greater than 0.05; hence, the two matched groups were not significantly different. This showed that the inside pixels were successfully matched with the outside pixels. A total of 108 PAs were assessed, and 5752 pixels were successfully matched from 2008 to 2018 (Table 4, Figure 12). The deforestation rates inside and outside PAs were 4.64% and 4.92%, respectively. The AE was 0.28%, supporting the effectiveness of PAs. The RE of PAs in Himalaya from 2008 to 2018 was 5.69%.

**Table 4.** Balance of control covariates from 2008 to 2018. Values are the mean (SD).

| Control Variables | Inside PAs (1) | Outside PAs (0) | *p*-Value |
| --- | --- | --- | --- |
| | *n* = 5752 | *n* = 5752 | |
| DistanceToRiver | 1844.50 (1286.48) | 1825.85 (1257.85) | 0.432 |
| DistanceToRoad | 5844.12 (5673.20) | 5644.36 (5449.80) | 0.054 |
| DistanceToSettlement | 30,474.01 (22,260.53) | 29,803.18 (20,496.75) | 0.093 |
| Elevation | 2248.87 (1553.54) | 2165.43 (1536.90) | 0.004 |
| Precipitation | 1356.38 (679.42) | 1355.79 (722.95) | 0.964 |
| Slope | 22.44 (13.68) | 22.33 (12.85) | 0.669 |
| Soil organic carbon | 4.07 (2.80) | 3.89 (3.17) | 0.001 |
| Temperature | 12.51 (9.27) | 13.02 (9.14) | 0.003 |

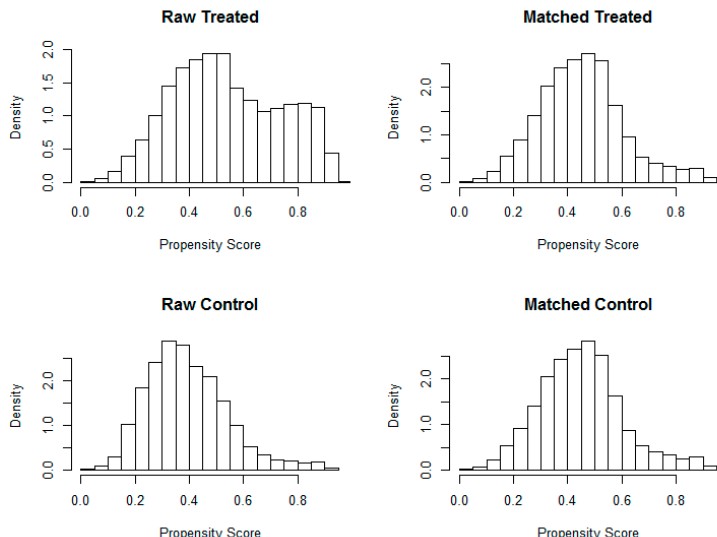

**Figure 12.** Distribution of propensity scores before and after PSM from 2008 to 2018. The Raw Control panel indicates the randomly selected pixels outside the PAs, and the Raw Treated panel indicates randomly selected pixels inside the PAs. The Matched Control panel denotes the matched pixels outside the PAs, while the Matched Treated panel denotes the matched pixels inside the PAs.

## 4. Discussion

### 4.1. Forest Cover and Fragmentaion Changes in Himalaya

Forests play a critical role in biodiversity conservation and local livelihoods in Himalaya [16], where forest degradation is regarded a major driver of biodiversity loss [108]. However, the relevant research on this topic is limited and mainly done at the regional scale, and the fourth and fifth reports of the Intergovernmental Panel on Climate Change (IPCC), which is located in Geneva, Switzerland, explicitly pointed to the Hindu Kush Himalaya (HKH) as a data deficit area [109,110]. Our study mapped high-accuracy forest cover data from 1998 to 2018 based on the Google Earth Engine, which reduced the uncertainty of estimates of changes in forest cover. By quantifying spatial–temporal changes of both forest cover and fragmentation from 1998 to 2018 in Himalaya, we discerned a net increase in Himalaya forest area of 250.94 km$^2$. Even so, the deforestation (which decreased by 2.2%) that occurred in the last 10 years almost entirely offset the reforestation (increased by 2.37%) in the first 10 years. Similar patterns were also observed in other Himalayan regions [3,111,112].

Compared with the global rate of forest loss estimated by Keenan et al. (2015), the Himalayan rate of forest loss was higher (0.22% vs. 0.12% per year); in a regional-level comparison, this number was lower (0.22% per year vs. the 0.34% per year for Southeast Asia). Nevertheless, substantial regional differences exist in Himalaya. In the temperate region of Western Himalaya, Pakistan, the average annual rate of deforestation from 1990 to 2013 was 0.16% [3], while in the Eastern Himalaya and Sikkim Himalaya in India it was much higher, at 0.7% [27]. We found an average annual rate of deforestation

in Eastern Himalaya from 1998 to 2018 of 0.39%, nearly double that estimated for the whole Himalaya (0.22%). Forest fragmentation was further analyzed from 1998 to 2018 in Himalaya, and there was a certain similarity in its changes in forest cover vis-à-vis forest fragmentation. The decrease of cores and the increase of patch, edge, and perforated forest cover types showed that the problems of forest fragmentation in Himalaya worsened from 1998 to 2018. Similar phenomena have been reported in regional areas of Himalaya [16,113]. In Eastern Himalaya, the large core was the dominant type, and it decreased by 4.36% in the last 10 years. Meanwhile, the patch, edge, and perforated increased by 1.45% in total. These results indicated that forest cover loss mainly occurred in Eastern Himalaya from 1998 to 2018. By analyzing the transfer process between forest types and non-forest areas, we found that changes to edges contributed most to the forest cover change. Studies have shown that forest cover changes have stronger linear relationships with edges [21], which indicated that they are more likely to unfold at the edges of forests. Forest fragmentation can pose a threat to ecosystems, causing habitat loss and harming the biodiversity conservation [20,114,115]. Nevertheless, research on this is scarce, coming mainly from Western and Central Himalaya [27]. Thus, our study could provide a certain reference of significance for forest conservation policy and planning in Eastern Himalaya.

Human activities, including agricultural expansion and road construction, were probably the main drivers of deforestation and forest fragmentation of Himalaya in the 20-year period studied here. Anthropogenic activities are generally seen as the main driver of forest cover change [116,117], and the influence of such activities has intensified remarkably in Himalaya during the last few hundred years, especially in its middle altitudinal zones [118]. Both elevation and slope can serve as indicators to evaluate the accessibility and, therefore, to determine the potential for human activities' intensity [27]. Land change has been greater in lowland areas in the tropics due to their flat landscapes that are more favorable for agriculture activities [119]. The 300–2500 m elevation range is known as the agro-ecological zone, consisting of a wide variety of production systems and crop cultivation [27]. Accordingly, there would be more intense human activities carried out in this range. In our study, over 55% of the remotely observed deforestation accrued under 2000 m elevation in each of the three study periods. For example, Munsi et al. (2008) [13] quantified land use and land cover change from 1976 to 2006 in Uttarakhand, and found that there was a 6% decline in the mid-elevation regions. Panta et al. [120] analyzed deforestation and forest degradation in Chitwan, a National Park in central Nepal, where forest losses were at quite a high rate at lower and middle elevations (110–1945 m). It also seems that the middle hills of the Himalaya have experienced forest deforestation for decades [121]; however, the issue here is that biological diversity often peaks at intermediate elevations [122]. In addition, road infrastructure is considered as another critical driver of forest fragmentation [123]. Recently, Mann et al. (2019) reported that, from 2000 to 2016, forest degradation has evidently increased along the roads dissecting the forest landscapes in Central Himalaya [50]. Similar findings indicated that forest degradation patterns in the Brazilian Amazon will be driven by the distributions of road networks [124,125]. Meanwhile, other studies forecast a 60% increase in road length in 2050 compared with 2010 and that 90% of this increase will occur in developing nations [126]. How to solve the contradiction between development and protection becomes an urgent issue.

### 4.2. Measurimg the Effectiveness of PAs in Himalaya

PAs were deemed as an effective way to mitigate forest degradation, yet the effectiveness of PAs is still a controversial matter. The inside–outside comparisons we used have been applied widely for assessing the effectiveness of PAs [127–129]. The problem with this kind of research is losing sight of inherent spatial heterogeneity in the landscape. Furthermore, the locations of PAs are not always random; PAs tend to be built in remote areas, where there is already relatively low human pressure [106], so external pressure would have been at a low level even without protection. In this study, we used PSM with eight control variables to reduce the influence of spatial heterogeneity on the effectiveness of PAs. Based on the successfully matched samples inside and outside PAs, the effectiveness of PAs in Himalaya could be robustly quantified. We found that even though forest cover of about 34% and

68% of PAs showed a decreasing trend in two study periods (1998–2008, 2008–2018), as a collective network, the PAs in Himalaya played their anticipated part in halting deforestation. From 1998 to 2008, the deforestation rate inside PAs was 3.3%, and this number was 3.76% outside PAs, which means that deforestation was avoided by 0.46% due to PAs' presence. During the period of 2008 to 2018, 4.64% and 4.92% deforestation rates were respectively observed inside and outside PAs; hence, this corresponded to 0.28% avoidance of deforestation of PAs. The effectiveness of PAs during the first period (1998–2008) seems to have outperformed the latter period (2008–2018). These results agree with the aforementioned forest cover changes from 1998 to 2018.

The differences in deforestation inside and outside PAs in Himalaya during the research period were relatively small when compared with findings from other regions. Cuenca et al. (2016) applied matching methods to assess the effectiveness of Ecuador's tropical Andean forest protected area system between 1990 and 2008, and found that PAs avoided approximately deforestation by 6% in tropical Andean landscapes [101]. The same method was implemented in Chile, for which it was estimated that deforestation was avoided by over 6% [130]. More recently, Zhao et al. (2019) evaluated the performance of nature reserves for forest protection in southwest China from 2000 to 2012; deforestation avoidance of natural reserves was measured, with reported estimates ranging from −15.79% to 14.73% and an average AE of 0.96% across all natural reserves in there [100]. Previous studies have shown that capacity and resources are correlated with the performance of PAs [131], in that insufficient protection inputs limit the management effectiveness in developing countries, which then influence the effectiveness of their PAs [132]. From this perspective, improving the level and scope of management may contribute to better-performing PAs.

All the same, the forest cover loss happening inside PAs merits attention. The ineffectiveness of protected area management may be one of the main causes for this [52]. The collection of non-timber forest products and fuel wood extraction in protected areas continues due to the lack of regulations and management [133]. Studies have shown that more strictly protected area categories tend to be more effective [134]; however, we found that even the Ia PA was unable to attain the best level of protection. Similar results have been observed in other studies [96,135]. Furthermore, the deforestation spillovers from protected areas are worth gauging and monitoring. Spillover effects refer to protection in one certain area influencing the non-protected adjoining areas, with potentially negative impacts for biodiversity [136,137]. The most forest cover lost was in the Chitwan Buffer Zone (VI), where it decreased by 13.04% from 1998 to 2018. However, Chitwan (II) showed the opposite trend, with an increase in forest cover of 1.75% from 1998 to 2018 (Figure S3). A similar phenomenon was also observed in other PAs, with numerous studies detecting such spillover effects in other PAs around the world. Jones et al. (2018) [138] found that anthropogenic pressures are on the rise outside the PAs, and when Bruggeman et al. (2018) [139] assessed PAs' effectiveness at resisting forest degradation, their findings suggested that they triggered a leakage of forest loss in surrounding areas.

Setting up PAs is regarded as a critical approach to conservation of biodiversity in situ. However, the trade-offs between biodiversity conservation and local people's livelihoods is also a subject worth considering [140]. The concept of buffer zones was introduced with the aim of alleviating the anthropogenic pressures upon PAs [141]. Research shows that buffer zones can reduce the damages caused, directly or indirectly, by humans [142,143]. Through statistical results of forest cover changes in PAs, we noticed that forest cover declines in the buffer zones (Figure S3). Taking the Chitwan National Park as an example, forest cover there increased by over 5% from 1998 to 2018, but forest cover in the buffer zone decreased by about 2% during the 20-year research period. Panta et al. (2008) [124] had analyzed the land cover change in Chitwan National Park, remarking that "the forests outside the protected area appear to be in poor condition with stagnant growth." The buffer zones endured the spillover effects caused by the establishment of Chitwan National Park, whose internal deforestation was thereby avoided. Building a buffer zone around a PA seems to be important and necessary.

*4.3. Uncertainties and Limitations*

Our study also has some limitations. First, the cloudy conditions and topographic shadows made the data acquisition and processing steps very difficult, though long time-series forest cover data can reveal more details behind the forest degradation dynamics. In this study, we used PSM to assess the effectiveness of PAs, but did not take natural disasters (e.g., fires, floods) into consideration, even if the study pointed out that forest fire causes forest degradation and change in landscape patterns in India [144]. Furthermore, our assessments of PAs are at the scale of the entire Himalaya, rather than on an individual scale. We assessed the effectiveness of all PAs pooled during the different study periods; hence, the effectiveness of a particular PA warrants discussion and investigation. Taking a regional scale approach also meant that the management level of PAs in different nations could hardly be taken into consideration. This may have led to overestimation or underestimation of the effectiveness of some PAs. Hence, we think that it is necessary to not only build long time-series forest cover datasets for the Himalaya, but also to assess the individual-level performance of its PAs.

## 5. Conclusions

This study aimed to quantify the spatiotemporal patterns of forest cover and fragmentation in Himalaya (based on the Google Earth Engine) and to further assess the effectiveness of PAs in halting deforestation over the past 20 years (1998 to 2018) by using a propensity score matching (PSM) method. This study revealed that the process of forest change increased between 1998 and 2008 and then decreased between 2008 and 2018. From 1998 to 2008, forest cover change (increased by 3175.77 km$^2$) in Eastern Himalaya (EH) contributed most to the forest cover increase of Himalaya, but from 2008 to 2018, 67% of the decrease also occurred there. Therefore, forest cover fluctuation was largely driven by dynamics in the EH. Further analysis indicated that over 55% of the deforestation and reforestation happened in lower-lying areas (<2000 m elevation) that are more accessible to human activities. Forest fragmentation analyses showed that large cores accounted for the largest proportion of all the forest fragmentation types, and these were mainly distributed in EH. The changes of edges contributed most to the forest fluctuation during the 20-year study period, suggesting that changes in forest cover were driven by dynamics at the edges of forest tracts.

Despite forest cover declining in about 56% of PAs from 1998 to 2018, the PAs in Himalaya effectively reduced deforestation by approximately 0.51% inside PAs. That means that the PAs avoided 0.51% of deforestation during the 20-year study period. Even so, some forest cover loss occurred inside PAs that was still noteworthy. These findings could inform forest conservation efforts and could help policymakers better understand spatiotemporal characteristics of forest cover changes in Himalaya during the past 20 years. The individual-level effectiveness of PAs deserves further study to assess the performance of each PA in the network to help improve their protection capabilities.

**Supplementary Materials:** The following are available online at http://www.mdpi.com/2071-1050/12/15/6123/s1, Figure S1: Forest cover map of Himalaya in 1998, 2008 and 2018, Figure S2: Deforestation and reforestation in three periods of Himalaya, Figure S3: Amplification map of forest cover change in Chitwan National Park, Table S1: Statistics of the 108 PAs in Himalaya. Name, type and built year were specified by NAME, IUCN_CAT and STATUS_YEAR, respectively, Table S2: The data source and resolution of eight control variables.

**Author Contributions:** L.L. (Linshan Liu) and Y.Z. designed the study. C.G., Q.C., and P.Z. conceived the research; C.G. studied and wrote the paper; P.Z. mapped the forest cover data and drew the figures; S.L. and L.L. (Lanhui Li) revised the paper and polished the language. Y.Z. revised the paper and contributed to the explanation of the results and the discussion. All authors have read and agreed to the published version of the manuscript.

**Funding:** This research was funded by the second Tibetan Plateau Scientific Expedition and Research Program, Grant No. 2019QZKK0603; Strategic Priority Research Program of Chinese Academy of Sciences, Grant No. XDA20040201; and National Natural Science Foundation of China (41761144081, 41671104).

**Acknowledgments:** We would like to express our special thanks to Xue Wu, Qionghuan Liu, and Binghua Zhang from the Institute of Geographic Sciences and Natural Resources Research, CAS, Xue Wu from Qinghai Normal University, Huamin Zhang and Beibei Wang from Jiangxi Normal University, and Popo Wu and Zi Wang from Institute of Botany, CAS for their great help in the field work. We are particularly grateful to Hua Zhang from Jiangxi Normal University for his timely help.

**Conflicts of Interest:** The authors declare no conflict of interest.

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
