# Peer review of "Forest Cover Change and the Effectiveness of Protected Areas in the Himalaya since 1998"

_sustainability, doi:10.3390/su12156123_

Round 1

Reviewer 1 Report

This is review of the manuscript “Forest cover change and the effectiveness of Protected Areas in Himalayas since 1998” by Changjun Gu submitted for publication in sustainability. I feel that the content of this MS is interesting, but the set question c) (Lines 86-87) is not well discussed. I recommend that this MS not be accepted without major revision.

Many abbreviations make it difficult to read overall.

Line 151: Replace “Vegetation” into “Water”.

Lines 258-259: This sentence is different from the result in Table 1. Is “1998-2018” not “1998-2008”?

Line 265: Replace “Table 1” into “Table 2”.

Line 271: Replace “100m” into “1000m”.

Line 272: Replace “Table 1” into “Table 2”.

Lines 281: The change in “small core” and “medium core” has a small absolute value and cannot be read from Figure 7.

Lines 315: The layout of “LC” and “MC” in Figure 9 should be upside down.

Line 342: There is one unnecessary horizontal rule in Table 3.

Lines 384-438: I think that Section “4.1” is mostly general citation with many references, and the main issue obtained from the data does not reach the reader.

Lines 465-467: I can’t really understand this sentence of “Building the buffer-zone around the PA seems to be important and necessary”.

Lines 469-470: In this study, can you numerically show how much the cloudy conditions and topographic shadows were limited by?

Lines 488-489: This sentence is the same expression as Lines 392-393.

Reviewer 2 Report

My comments are in the enclosed PDF file.

Round 2

Reviewer 1 Report

The manuscript has been revised well. I think this manuscript will be acceptable after some corrections have been done.

Line 49: Replace “Km2” into “km2”. Check carefully for similar misspellings.

The manuscript I received contains two different tables as Table 1, so make sure they are correct.

Reviewer 2 Report

There are still mistakes in the writing that need to be corrected. The authors have made efforts, but all the mistakes have not been corrected.
